# Development and biophysical characterization of a humanized FSH–blocking monoclonal antibody therapeutic formulated at an ultra-high concentration

Satish Rojekar[1]*, Anusha R Pallapati[1], Judit Gimenez-Roig[1], Funda Korkmaz[1], Farhath Sultana[1], Damini Sant[1], Clement M Haeck[2], Anne Macdonald[1], Se-Min Kim[1], Clifford J Rosen[3], Orly Barak[1], Marcia Meseck[1], John Caminis[1], Daria Lizneva[1], Tony Yuen[1], Mone Zaidi[1]*

[1]Center for Translational Medicine and Pharmacology, Icahn School of Medicine at Mount Sinai, New York, United States; [2]Center for Biomedical Research, Population Council, New York, United States; [3]Maine Medical Center Research Institute, Scarborough, United States

*For correspondence:
satish.rojekar@mssm.edu (SR);
mone.zaidi@mountsinai.org (MZ)

**Abstract** Highly concentrated antibody formulations are oftentimes required for subcutaneous, self-administered biologics. Here, we report the development of a unique formulation for our first-in-class FSH-blocking humanized antibody, MS-Hu6, which we propose to move to the clinic for osteoporosis, obesity, and Alzheimer's disease. The studies were carried out using our Good Laboratory Practice (GLP) platform, compliant with the Code of Federal Regulations (Title 21, Part 58). We first used protein thermal shift, size exclusion chromatography, and dynamic light scattering to examine MS-Hu6 concentrations between 1 and 100 mg/mL. We found that thermal, monomeric, and colloidal stability of formulated MS-Hu6 was maintained at a concentration of 100 mg/mL. The addition of the antioxidant L-methionine and chelating agent disodium EDTA improved the formulation's long-term colloidal and thermal stability. Thermal stability was further confirmed by Nano differential scanning calorimetry (DSC). Physiochemical properties of formulated MS-Hu6, including viscosity, turbidity, and clarity, confirmed with acceptable industry standards. That the structural integrity of MS-Hu6 in formulation was maintained was proven through Circular Dichroism (CD) and Fourier Transform Infrared (FTIR) Spectroscopy. Three rapid freeze–thaw cycles at –80 °C/25 °C or –80 °C/37 °C further revealed excellent thermal and colloidal stability. Furthermore, formulated MS-Hu6, particularly its Fab domain, displayed thermal and monomeric storage stability for more than 90 days at 4°C and 25°C. Finally, the unfolding temperature ($T_m$) for formulated MS-Hu6 increased by >4.80 °C upon binding to recombinant FSH, indicating highly specific ligand binding. Overall, we document the feasibility of developing a stable, manufacturable and transportable MS-Hu6 formulation at a ultra-high concentration at industry standards. The study should become a resource for developing biologic formulations in academic medical centers.

## Editor's evaluation

This development of a highly concentrated and potentially clinically valuable antibody formulation for MS-Hu6, a first-in-class FSH-blocking humanized antibody is of potential translational importance in the management of osteoporosis, obesity, and Alzheimer's disease. The meticulous methodology

is thorough and compelling in its range of techniques examining the stability and physiochemical properties of the formulated MS-Hu6.

## Introduction

Biotherapeutics have transformed the treatment of chronic inflammatory diseases and cancers (*Lu et al., 2020*; *Passot et al., 2016*; *Wang et al., 2021*), with monoclonal antibodies witnessing the most rapid growth because of their specificity and adaptability (*An, 2010*; *Shire et al., 2004*; *Whitaker et al., 2017*). As monoclonal antibodies, particularly blocking antibodies, are generally of low affinity, and therefore must be administered at relatively high doses to elicit therapeutic responses (*Whitaker et al., 2017*; *Elbakri et al., 2010*). Mostly given subcutaneously (*Wang et al., 2021*), they are utilized at a maximally allowable volume of 1.5 mL. It has therefore become critical to develop formulations of monoclonal antibodies at ultra-high concentrations (*Wang et al., 2021*; *Storz, 2016*). These formulations also allow single-dose self-administration using auto-injection devices, reduced dosing frequency, and improved patient compliance (*Wang et al., 2021*; *BioProcess International, 2014*).

However, ultra-high concentration monoclonal antibody formulations are associated with physical and chemical instabilities depending on the physicochemical properties of the specific monoclonal antibody (*Bramham et al., 2021*; *Jiskoot et al., 2022*; *Klich et al., 2023*). Notably, due to the higher concentrations, these formulations may be very viscous or gelatinous, leading to increased tissue back pressure and pain at the injection site (*Whitaker et al., 2017*; *Jezek et al., 2011*). Manufacturing challenges for highly concentrated formulations also include difficulty in pumping due to high shear stress, clogging of the membrane, high back pressure, and aggregation during the manufacturing (*Thomas et al., 1979*; *Bee et al., 2009*; *Hung et al., 2016*; *Goswami et al., 2013*).

Despite these obstacles, 34 out of 103 monoclonal antibody therapeutics approved by the U.S. Food and Drug Administration (FDA), that constitute almost one third of all approved biotherapeutics, have been developed with high protein concentrations (≥100 mg/mL); 76% of the approved formulations are administered by the subcutaneous route (*Wang et al., 2021*; *Storz, 2016*). The very first ultra–high concentration monoclonal antibody formulation, Palivizumab (Synagis, 100 mg/mL), was approved in 1998 (*Geskey et al., 2007*), followed by others including Alirocumab (150 mg/mL, Sanofi–Regeneron), Cosentyx (150 mg/mL, Novartis), Actemra (180 mg/mL, Genentech), and Belimumab (200 mg/mL, GlaxoSmithKline) (*Lu et al., 2020*; *Whitaker et al., 2017*; *Baldo, 2016*).

The main issue with highly concentrated formulations is protein–protein interactions (*Shire et al., 2004*; *BioProcess International, 2014*) resulting from high viscosity and a crowded environment (*Whitaker et al., 2017*; *Buck et al., 2015*; *Arora et al., 2015*). Such interactions are reversible or irreversible, and may cause undesirable liquid–liquid separation, protein aggregation, or opalescence (*Whitaker et al., 2017*; *Ratanji et al., 2016*). The effects may also be caused by a variety of transient interactions in the solution, including hydrophobic, electrostatic, dipole–dipole, hydrogen bonding (inter and intra), or van der Waals interactions (*Esfandiary et al., 2015*; *Connolly et al., 2012*; *Singh et al., 2014*) *via* Fab-Fab (*Buck et al., 2015*) or, in rare instances, Fab-Fc (*Arora et al., 2016*) interactions between antibodies.

Proteins generally undergo aggregation using native monomers, denatured proteins, and preexisting aggregates (*Goswami et al., 2013*; *Narhi et al., 2012*; *Pham and Meng, 2020*). Native monomers self-associate into the oligomers through transient interactions and regain their native structure—most such associations result in reversible aggregates. However, upon stressing, and over time, monomers can associate to form multimers, resulting in irreversible (insoluble) aggregates (*Ratanji et al., 2016*; *Pham and Meng, 2020*). While 5–10% soluble aggregates are acceptable for biologic therapeutics, insoluble aggregates must be limited for stable formulations (*Pham and Meng, 2020*; *Wang et al., 2012*). The US Pharmacopeia (USP) and the FDA have set guidelines to control aggregates in a final biologic formulation (*U.S Food and Drug Administration, 2021*; *USP, 2012*). For example, sub–visible particles with a hydrodynamic radius ≤50–100 μm are considered optimal (*Ratanji et al., 2016*; *Wang et al., 2012*). Hence, *per* the FDA, the acceptable limit of subvisible particulate matter in a container of ≤100 mL should be 6000 particles ≥10 μm and 600 particles ≥25 μm (*Ratanji et al., 2016*; *Pham and Meng, 2020*; *U.S Food and Drug Administration, 2021*; *Pham and Meng, 2020*). Particle sizes ≤10 μm can cause vascular occlusion.

We have reported the development of a first-of-its-kind humanized FSH-blocking antibody, MS-Hu6, based on studies that have established direct actions of FSH in promoting bone loss, body fat accrual, and neurodegeneration. MS-Hu6 reduces body fat, induces thermogenic adipose tissue, improves bone density, and prevents cognitive decline in mouse models (*Gera et al., 2020*; *Zaidi et al., 2018*; *Liu et al., 2017*; *Sun et al., 2006*; *Zhu et al., 2012*; *Gera et al., 2022*; *Zaidi et al., 2023*; *Xiong et al., 2022*). MS-Hu6 was humanized from the parent murine antibody clone, Hf2, generated against a short 13-amino-acid-long epitope of the receptor-binding sequence of FSHβ (*Ji et al., 2018*). We have shown that MS-Hu6 binds to FSHβ and blocks its interaction with the FSH receptor (FSHR) with an affinity of 7.5 nM, comparable to trastuzumab ($K_D$ ~5 nM, Herceptin; *Gera et al., 2022*). We have also confirmed the stability of MS-Hu6 and its binding to FSH using a protein thermal shift assay (also known as differential scanning fluorimetry; *Gera et al., 2022*; *Sant et al., 2023*).

We have recently reported a low-concentration formulation for MS-Hu6 (*Sant et al., 2023*). A high-throughput protein thermal shift assay was used to screen different formulation parameters, including pH, buffer, salt, surfactant, and sugar. This optimized formulation consisted of 0.15 mg/mL of MS-Hu6, 20 mM phosphate buffer (pH 6.2), 1 mM sodium chloride, 0.001% w/v Tween 20 and 260 mM sucrose. Here, we have formulated an ultra–high concentration of MS-Hu6 at 100 mg/mL. We have fully characterized the effect of antibody concentration on formulation stability using the protein thermal shift assay, dynamic light scattering (DLS), and size exclusion chromatography. For long-term and optimum stability, we screened antioxidant and chelating agent concentrations using the protein thermal shift assay and DLS. In addition, this optimized formulation was also examined for viscosity, turbidity, and clarity; thermal stability using differential scanning calorimetry (DSC), freeze–thaw studies, and accelerated stability; and importantly, FSH binding. Structural integrity was also evaluated using FTIR and CD spectroscopy. These studies were carried out under Good Laboratory Practice (GLP) standards *per* Code of Federal Regulations (Title 21, Part 58), applied perhaps for the first time, in this context, in an academic medical center.

## Results

### Optimizing the concentration of MS-Hu6 formulations

The ultra–high concentration of formulated MS-Hu6 was optimized using protein thermal shift for thermostability, size exclusion chromatography (SEC) for monomeric stability, and dynamic light scattering (DLS) for colloidal stability. Thermal shift determines a protein's unfolding (melting) temperature ($T_m$), which is the temperature at which 50% of molecules are denatured (*Reaction Biology, 2023*). Denaturation is observed by the increase in fluorescence of SYPRO Orange, which binds to hydrophobic residues that are exposed as globular proteins unfold (*Gera et al., 2022*). Binding of ligand to the target protein results in increased stability and a higher $T_m$. $T_m$ values that are higher (or more stable) indicate overall thermostability. To optimize the formulation, we increased MS-Hu6 concentration from 0.2 to 100 mg/mL (*Figure 1A*). $T_m$ values for the Fc and Fab domains of MS-Hu6 in PBS at 1 mg/mL were 69.17 ± 0.02 °C and 79.64 ± 0.02 °C, respectively. Increases $T_m$ values compared with MS-Hu6 in PBS were noted progressively up to 10 and 20 mg/mL for the Fab and Fc regions of formulated MS-Hu6. However, at higher concentrations (50 and 100 mg/mL), while $T_m$ values dropped somewhat for the MS-Hu6 formulation compared with MS-Hu6 in PBS, the difference (or $\Delta T_m$) did not exceed 1 °C—which provided assurance of thermostability even at 100 mg/mL.

We further used SEC to study monomeric stability of formulated MS-Hu6 based on molecular size. The percentage of monomer (peak 2) and dimer (peak 1) areas were monitored. *Figure 1B* shows that MS-Hu6 in PBS or in formulation exists mainly in its monomeric form (>99%) with a dimer area of <1%; the latter is within acceptable industry standards of <3–5%.

We examined colloidal stability using DLS, a well-recognized, quick, and sensitive method to determine particle size and particle-size distribution. The method is based on light scattering due to Brownian motion. While maintaining the original structure of proteins, DLS measures the hydrodynamic diameter (rh) and polydispersity index (PDI). The samples were diluted with respective buffers (PBS or Formulation Buffer) prior to DLS. Data were collected in volume percent, with hydrodynamic radius (rh) presented for the prominent and smaller peaks (*Figure 1C*). Particle size data were reported in *Z*–average values.

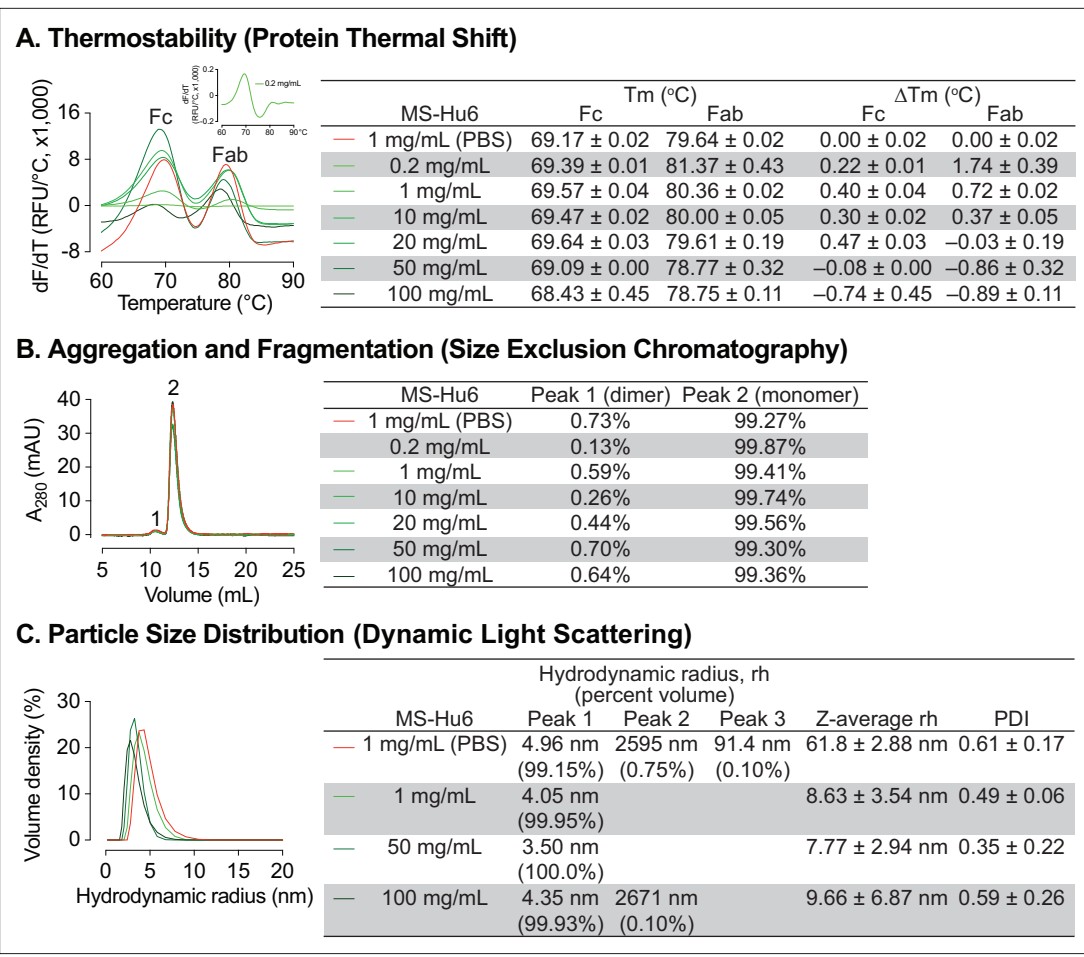

**Figure 1.** Thermal, monomeric and colloidal stability of formulated MS-Hu6 at increasing concentrations. (**A**) Protein thermal shift assay confirmed thermostability of formulated MS-Hu6 between 0.2 mg/mL and 100 mg/mL. Compared with MS-Hu6 in PBS, the difference in melting temperature ($\Delta T_m$) of the constant (Fc) and antigen-binding fragments (Fab) of formulated MS-Hu6 was <-1°C at all concentrations. (**B**) Size exclusion chromatography confirmed the monomeric nature of formulated MS-Hu6 between 1 and 100 mg/mL. Monomer loss was <1% in all concentrations. (**C**) Particle size distribution using dynamic light scattering yielded hydrodynamic radius (rh) and polydispersity index (PDI) of formulated MS-Hu6. Excellent colloidal stability (rh <10 nm, PDI <1) was found at concentrations from 1 to 100 mg/mL.

The online version of this article includes the following source data for figure 1:

**Source data 1.** Source data for *Figure 1*.

The two MS-Hu6 (in PBS) samples displayed a prominent peak (volume %~99%) with an average rh of 4.96 nm, which fell within the industry standard (rh <10 nm). However, there were two aggregated peaks with average volume percentages of <1%, but with an average rh of 91.4 and 2595 nm, respectively. The multiplicity of such peaks resulted in a high PDI, indicating varying particle size (*Figure 1C*). In contrast, MS-Hu6 formulations containing 1 and 50 mg/mL MS-Hu6 displayed 99–100 volume % in a single peak, with average rh values of ~4 nm. In the 100 mg/mL formulation, however, a negligible peak (0.1% in sample 3) was associated with a large rh of 2671 nm, whereas the other samples (1 and 2) displayed single peaks with an average rh of 3.68 nm. This resulted in lower PDI values for 1 and 50 mg/mL, with a similar PDI for the 100 mg/mL formulation to that of MS-Hu6 in PBS. Nonetheless, the PDI values in all cases were <1, which is the accepted cut-off. Large particles with a high rh values can form as soluble reversible aggregates, which are well accepted in biotherapeutics development at levels <10%. Thus, in all, the 100 mg/mL MS-Hu6 formulation showed an acceptable rh and PDI, suggesting that the monomeric nature of MS-Hu6 was maintained at higher concentrations.

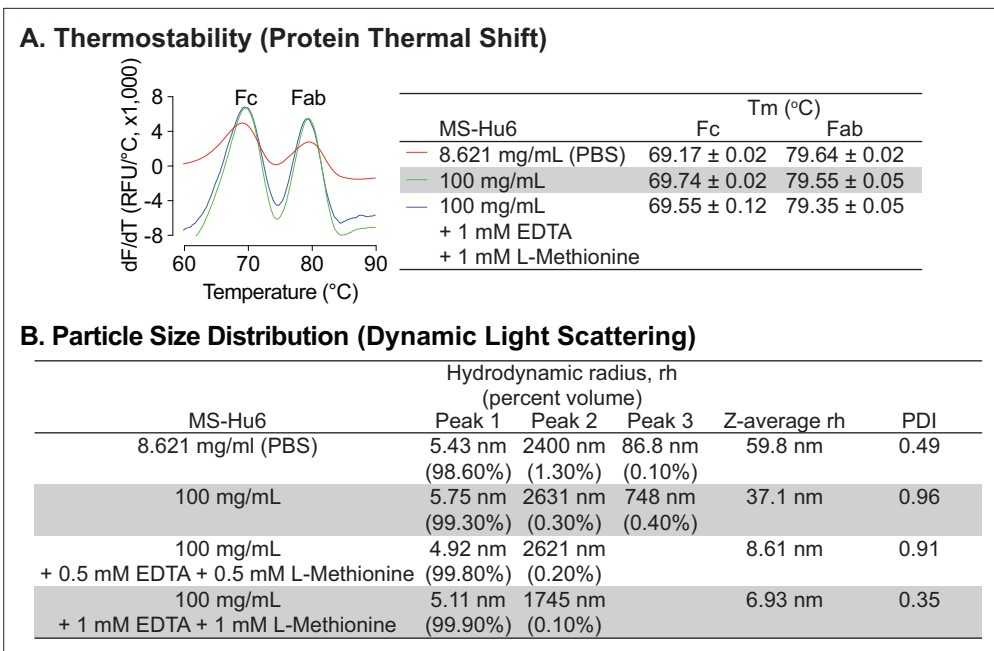

**Figure 2.** Thermal and colloidal stability of formulated MS-Hu6 is maintained in the presence of L-methionine and EDTA. To improve long-term storage stability, an antioxidant L-methionine and a chelating agent EDTA were added to formulated MS-Hu6. Addition of L-methionine and EDTA to formulated MS-Hu6 did not alter its melting temperatures ($T_m$) in the protein thermal shift assay (**A**) or the hydrodynamic radius (rh or $Z$–average rh) or polydispersity index (PDI) in dynamic light scattering (**B**).

The online version of this article includes the following source data for figure 2:

**Source data 1.** Source data for *Figure 2*.

## Optimization of antioxidant and chelating agent

Due to the presence of trace elements and to avoid catalytic oxidation during manufacturing, disodium EDTA was tested as an excipient for formulated MS-Hu6. Furthermore, given that there are two methionine residues in the Fc domain, namely Met33($C_H$3) and Met209($C_H$3), we opted also to use L-methionine to avoid methionine oxidation during manufacturing and storage. We examined the effect of disodium–EDTA and L-methionine on the stability of the MS-Hu6 formulation using protein thermal shift and DLS. Protein thermal shift showed that compared with MS-Hu6 (8.621 mg/mL) in PBS, formulated MS-Hu6 (100 mg/mL) plus 1 mM EDTA and 1 mM L-methionine showed slight increased $\Delta T_m$ value for the Fc domain of 0.38 °C (*Figure 2A*). This suggested that methionine oxidation and catalytic oxidation were being prevented to enhance the thermostability of MS-Hu6. In contrast, there was no significant $T_m$ shift in the Fab domain upon EDTA or L-methionine addition.

We next examined the effect of disodium-EDTA and L-methionine on the colloidal stability of formulated MS-hu6 using DLS, with data collection in terms of % volume, rh values for the main and minor peaks, and $Z$–average rh values (*Figure 2B*). MS-Hu6 at 8.621 mg/mL in PBS displayed a major peak (98.6% volume) with an average rh of 5.43 nm, which fell within the industry standard (rh <10 nm). However, there were two minor aggregated peaks with <1.5% total volume, but with an average rh of 86.8 and 2400 nm, respectively. In contrast, formulated MS-Hu6 (100 mg/mL) displayed a main peak at a rh value of 5.75 nm with a higher, 99.3% volume; this size was consistent with the reported literature (*Akbas et al., 2018*). There were two minor peaks with <0.5% volume and rh values of 748 and 2631 nm, respectively. The addition of disodium-EDTA and L-methionine to the formulation, at 0.5 mM or 1 mM each, both within the IIG limit (*Zhou et al., 2012*; *Rayaprolu et al., 2018*; *Stadtman, 1990*), reduced the volume of the aggregated peaks and resulted in main peaks with near–100% volume, respectively. In all cases, PDI remained <1. These data suggested that disodium-EDTA and L-methionine together improved colloidal stability, so that MS-Hu6 in the formulation existed primarily as a monomer.

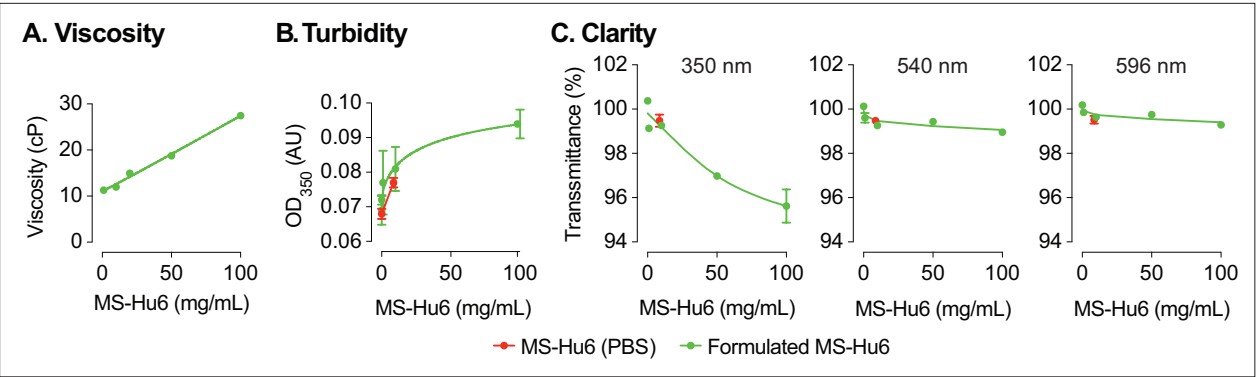

**Figure 3.** Evaluation of viscosity, turbidity, and clarity of the formulated MS-Hu6 at different concentrations. Formulated MS-Hu6 showed a concentration-dependent linear increase in viscosity within the acceptable industry standard (<50 centipoise, cP) (**A**). Formulations were also evaluated for turbidity (opalescence) at 350 nm. Low absorbance (AU, absorbance unit) was noted at lower concentrations; however, there was a concentration-dependent increase in absorbance at 100 mg/mL formulated MS-Hu6 (**B**). Formulated MS-Hu6 evaluated for clarity at wavelengths of 350, 540, and 595 nm demonstrated that the formulation is clear and transparent even at higher concentrations (100 mg/mL) that correlated well with the turbidity data (**C**).

The online version of this article includes the following source data for figure 3:

**Source data 1.** Source data for *Figure 3*.

## Evaluation of viscosity, turbidity, and clarity

We tested the effect of five concentrations of formulated MS-Hu6 (1, 10, 20, 50 and 100 mg/mL) on the dynamic viscosity, a surrogate measure for syringeability. Compared with water (3 cP) and formulation buffer (5.4 cP), formulated MS-Hu6 showed a concentration–dependent linear increase in viscosity (11.3, 12, 15, 18.8 and 27.5 cP, respectively; *Figure 3A*). The increase in viscosity likely resulted from the crowding of MS-Hu6 molecules that is expected at high antibody concentrations (*Raut and Kalonia, 2016*). However, importantly, the viscosity of the formulation at all MS-Hu6 concentrations were within the acceptable industry standard (<50 cP).

We also assessed turbidity (opalescence) of various concentrations of MS-Hu6 (in formulation at 0.2, 1, 10, and 100 mg/mL) using a microplate reader at 350 nm, as well as clarity at 350, 540, and 595 nm. Comparisons were made against MS-Hu6 in PBS (8.6251 mg/mL), PBS and formulation buffer—the latter showed very low absorbance at 0.068 (±0.002), 0.072 (±0.001), and 0.077 (±0.001), respectively (*Figure 3B*). There was a concentration-dependent increase in absorbance, with a flattening at 100 mg/mL MS-Hu6 in formulation. The increase at 100 mg/mL is likely due to antibody crowding in solution and formation of reversible aggregates; however, the latter phenomenon is not consequential in biopharmaceutical product development.

Finally, there were also no differences in clarity at any wavelength except a minimal drop with formulated MS-Hu6 at 100 mg/mL (*Figure 3C*). In all, the percent transmittance for each concentration was more than >99% at the three wavelengths, suggesting that the formulations are clear between 0.2 and 50 mg/mL and slightly opalescent at 100 mg/mL. All concentrations fall into the acceptable clarity categories of Ref I and II [3–6 Nephelometric Turbidity Units] (*European Pharmacopoeia, 2017*).

## Assessment of structural integrity

An alteration in the secondary structure or conformation of any protein indicates inactivation or denaturation. We used Circular Dichroism (CD) spectroscopy to evaluate the secondary structure of formulated MS-Hu6. As shown in *Figure 4A* and through mathematical derivations of ellipticity (millidegrees) and mean residue ellipticity, analysis in the far UV region (190–240 nm), revealed that, as with other IgG molecules, the predominant secondary structure in MS-Hu6 was regular β-sheets and unordered/random coils (*Kelly et al., 2005*; *Moreno et al., 2016*; *Vermeer et al., 1998*). Importantly, there was no alteration in any parameter in formulated MS-Hu6 *versus* MS-Hu6 in PBS (0.5 mg/mL).

However, since antibodies can be used for CD spectroscopy only at low concentrations, in our case at 0.5 mg/mL. We utilized Fourier-transform infrared (FTIR) spectroscopy to further study the

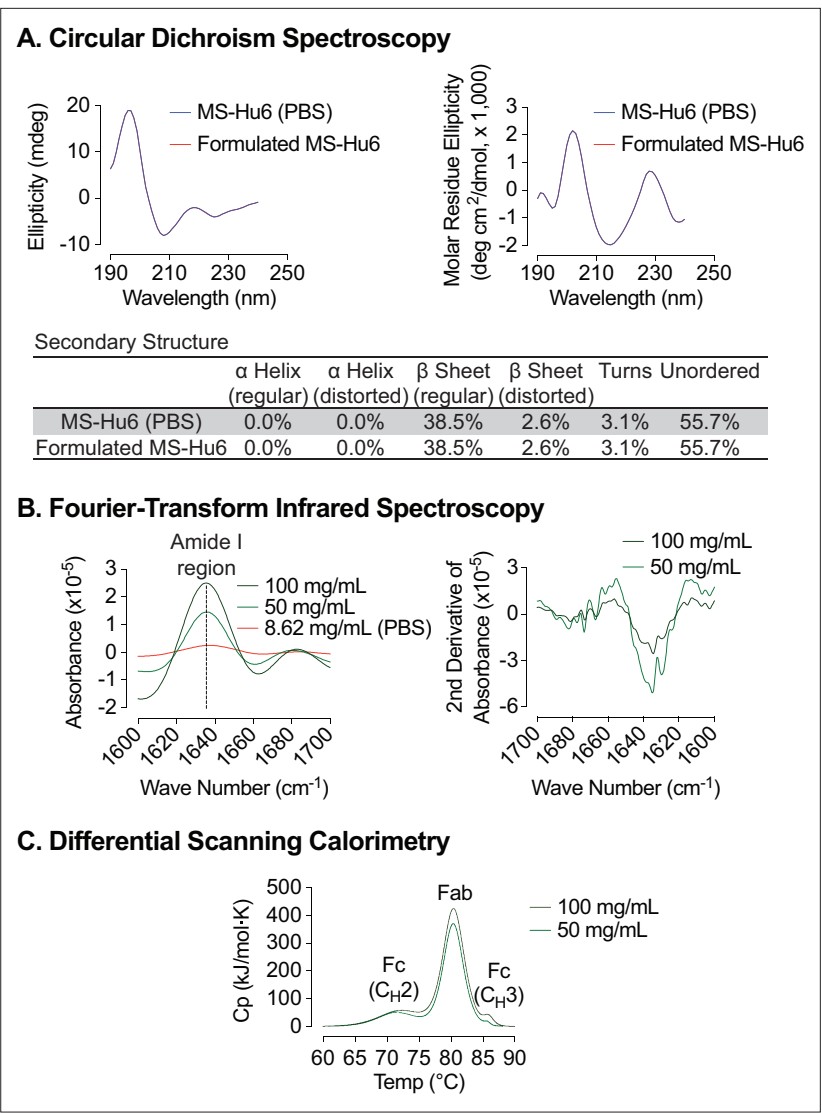

**Figure 4.** Biophysical characterization of formulated MS-Hu6. Circular dichroism (CD) spectroscopy evaluated the secondary structure of formulated MS-Hu6. Analysis in the far UV region (190–240 nm) revealed that the predominant secondary structure in MS-Hu6 was regular β-sheets and unordered/random coils (**A**). Secondary structure was also confirmed at higher formulation concentrations (50 and 100 mg/mL) using Fourier–transform infrared (FTIR) spectroscopy. The amide I band peak at 1637 cm⁻¹ (intra–molecular β-sheets) and the random coil (1642–1657 cm⁻¹) did not shift, confirming maintenance of the native conformation in formulation (**B**). Thermostability was further confirmed using nano differential scanning calorimetry (Nano DSC). MS-Hu6 concentrations of 50 and 100 mg/ml had comparable $T_m$s, indicating that the overall structure is conformationally and thermally stable at high concentrations (**C**).

The online version of this article includes the following source data for figure 4:

**Source data 1.** Source data for *Figure 4*.

structure of formulated MS-Hu6 at 50 and 100 mg/mL compared with MS-Hu6 in PBS. The amide I band peak at 1637 cm⁻¹, a surrogate for intra-molecular β-sheets, and the random coil structure peak at 1642–1657 cm⁻¹ did not shift, confirming maintenance of the antibody's native conformation in formulation (*Figure 4B*).

We finally employed nano differential scanning calorimetry (Nano DSC) to determine the protein unfolding temperatures ($T_m$s). MS-Hu6 at 50 and 100 mg/ml were diluted to 5 mg/mL in formulation buffer. MS-Hu6 displayed multi–domain transitions, with the first transition representing unfolding of $C_H2$ (Fc) domain ($T_m$ >71 °C), the second representing Fab unfolding ($T_m$ >80.30 °C), and a third $C_H3$

(Fc) domain unfolding ($T_m$ >85 °C) (*Figure 4C*). Both samples had comparable $T_m$s, indicating that the overall structure is conformationally and thermally stable at high concentrations. Furthermore, the heat capacity required for the $C_H3$ unfolding was ~two fold greater than Fab unfolding, and 7.73–8.52-fold greater than the $C_H2$ transition. This latter finding is consistent with what is observed for most IgG antibodies (*Ionescu et al., 2008*; *Ahrer et al., 2006*; *Garber and Demarest, 2007*).

## Rapid freeze–thaw stability evaluation

Protein instabilities, such as aggregation, are of major concern during biopharmaceutical development. Aggregation and denaturation can both take place during several phases of the product's lifecycle, including freeze–thaw, production, shipping, and storage. We carried out a formal rapid freeze–thaw (F/T) study on our MS-Hu6 formulation (100 mg/mL) at –80 °C/25 °C and –80 °C/37 °C, over three cycles, to confirm stability at extreme conditions. Samples were analyzed using protein thermal shift (PTS) and dynamic light scattering (DLS).

Baseline $T_m$s for formulated MS-Hu6 (100 mg/mL) without F/T were 69.55 ± 0.12°C and 79.35 ± 0.05°C for Fc and Fab, respectively (*Figure 5A*). Samples were stored at –80 °C and thawed to 25 °C for three cycles (I, II and III). Calculations of $\Delta T_m$ values for the both Fc and Fab domains, with each F/T cycle compared with baseline revealed no significant difference between each cycle and baseline. In another experimental set, samples were stored at –80 °C and thawed to 37 °C, over three cycles. While the Fc domain remained stable, there was a small, <1 °C, decrease in Fab $T_m$ compared with samples without F/T. The results demonstrate that formulated MS-Hu6 is stable under extreme stress and that there is minimal or no aggregation or denaturation.

All rapid F/T samples were further evaluated by DLS to determine their colloidal stability. For three cycles, the major volume peak (>99%) rh was found to be between 3–7 nm (*Figure 5B*). The Z–average rh for both experimental sets for all cycles remained <10 nm (except cycle III of –80/25 °C), with PDIs being <1. The small increase in PDI in cycle III (–80/37 °C) is likely due to the formation of reversible aggregates resulting from Fc–fc, Fc–Fab, or air-liquid interface interactions and/or reduced pH during buffer crystallization (*Raut and Kalonia, 2016*; *Kaur, 2021*; *Le Guyader et al., 2020*; *Torrente-López et al., 2022*). Nonetheless, the data are in agreement with findings for commercialized monoclonal antibody products, and demonstrate no or minimal reversible aggregation (<0.4%, for –80/25 °C, acceptable limit <5–10%). The formulation's excellent stability meets industry standards for product development and may improve transportability and manufacturability of MS-Hu6.

## Storage stability under accelerated conditions

Formulated MS-Hu6 (100 mg/mL) was evaluated for storage stability at 4°C and 25°C for 15, 30, 60, or 90 days. Samples were tested for thermostability using protein thermal shift and for monomer loss using size exclusion chromatography (SEC). At 4 °C, there was no appreciable $T_m$ shift for the Fab domain compared to day 0 (*Figure 6A*). However, for the Fc domain, a very negligible left–shift of 0.34 °C was noted at 90 days, but this was within the acceptable cut–off of 1 °C. When incubated at 25 °C, there is a small left–shift in the $T_m$s of both the Fab and Fc domains at day 15 ($\Delta T_m$ 0.92°C and 0.33°C, respectively), followed by relative good stability up to 90 days for Fab. However, during the last 90 days of incubation, there is a loss of thermostability for the Fc domain with a $\Delta T_m$ 1.51 °C. This data shows that formulated MS-Hu6 is thermostable at 4 °C (refrigerated conditions) and has excellent Fab stability in accelerated conditions at 25 °C for at least 90 days. These relatively stable $T_m$s likely arise from the stabilizing effects of the salt, stabilizer and sugar utilized in the optimized formulation.

We also assessed monomer loss by SEC. Samples stored at 4 °C retained 99.36% of monomer area at baseline (*Figure 6B*). After 90 days at 4 °C, all three batches retained an average monomeric area of 99.38 ± 0.18%, with monomer loss <1% (acceptable limit for active structure loss <5%) (*Whitaker et al., 2017*; *NHS, 2017*; *Wang et al., 2007*). These findings show that MS-Hu6 in formulation (100 mg/mL) maintained monomeric stability for more than 90 days. The formation of soluble reversible aggregates, normally as a result of antibody crowding in solution or weak Fc–Fc, Fc–Fab, Fab–Fab, hydrophobic or air–liquid interactions was almost non–existent, consistent with other robust IgG formulations. Moreover, no fragmentation was observed on SEC in any samples at 4 °C for 90 days.

For samples stored at 25 °C for 90 days, the average monomer area retained was 96.84 ± 1.22% for three batches, and monomer loss was 3.16% on average (*Figure 6B*). However, two of the samples (batches 1 and 2) lost ~4% monomer within the acceptable limit of <5%, while one sample retained

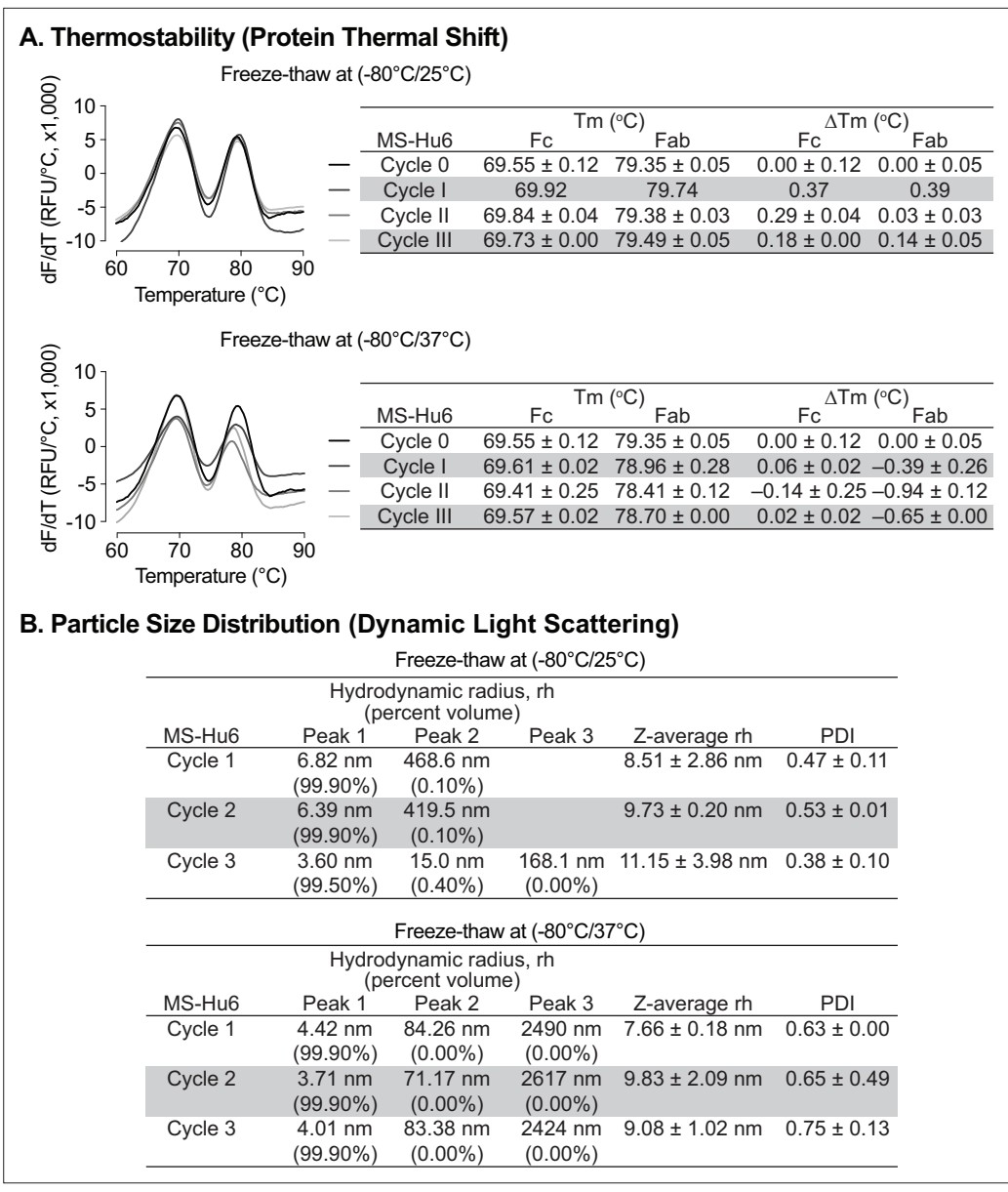

**Figure 5.** Evaluation of freeze-thaw stability of the formulated MS-Hu6 (100 mg/mL) at −80/25 °C and −80 °C/37 °C for three cycles. (**A**) Samples were analyzed using protein thermal shift for thermostability. After storage at −80/25 °C, $\Delta T_m$ values for both Fc and Fab domains revealed no significant difference between each cycle and baseline. However, for samples stored at −80 °C/37 °C (three cycles), while the Fc domain remained stable, there was a small change in Fab $\Delta T_m$ (within acceptable limit of 1 °C). (**B**) Samples were also analyzed using dynamic light scattering (DLS) for colloidal stability. The major peak (Peak 1) hydrodynamic radius (rh) was found to be between 3 and 7 nm at both storage conditions (−80/25 °C and −80 °C/37 °C). PDI—polydispersity index. The data suggest no or minimal reversible aggregation (<0.5%; acceptable limit: 5–10%).

The online version of this article includes the following source data for figure 5:

**Source data 1.** Source data for *Figure 5*.

99.29% monomer. This increased monomer loss at 90 days is likely due to the formation of soluble high molecular weight species (>150 kDa); these species are not considered critical in biopharmaceutical development due to their reversible and transient nature. In all, formulated MS-Hu6 at 4°C and 25°C for 90 days showed acceptable stability and retained maximum monomeric area in accordance with set standards.

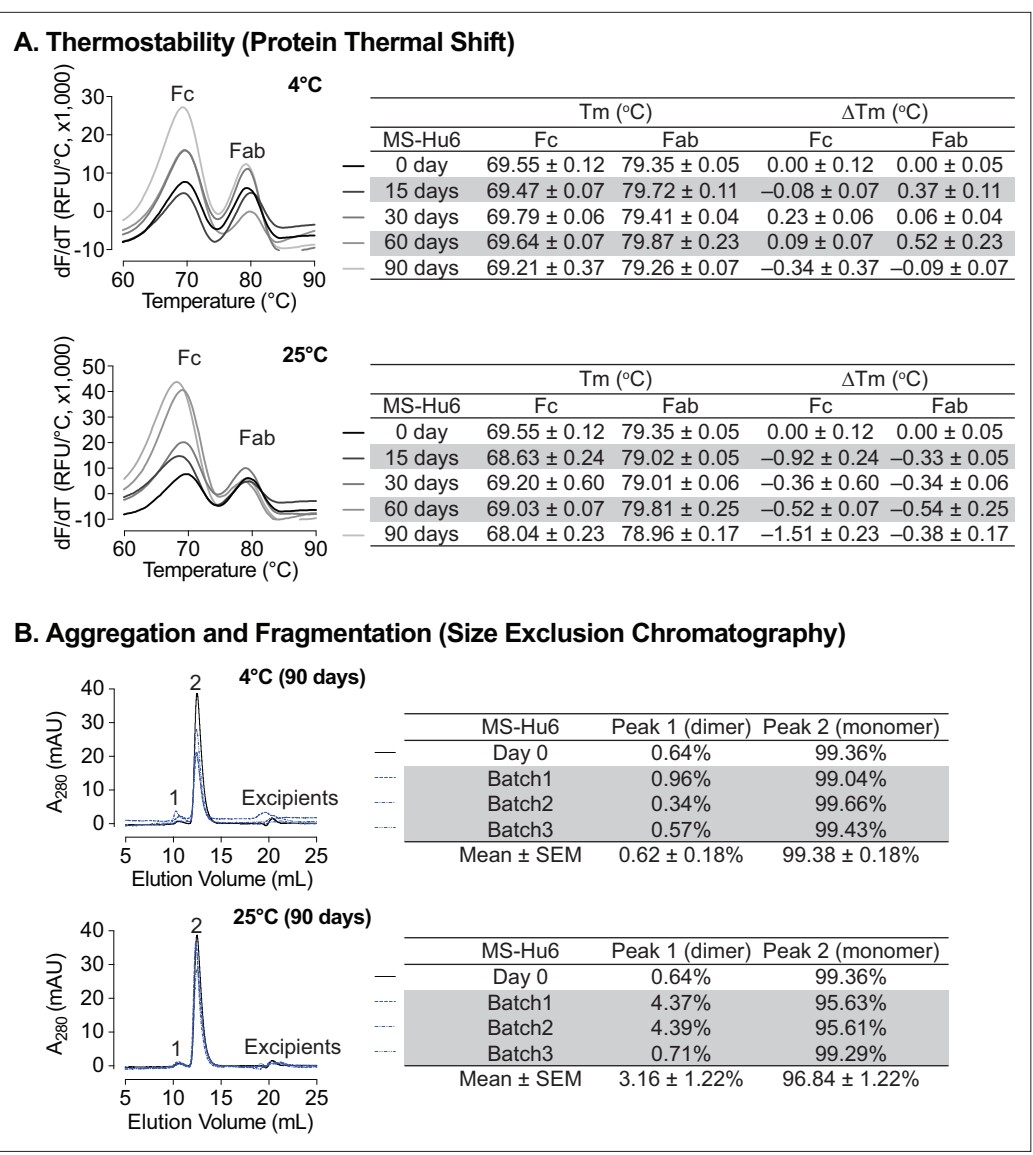

**Figure 6.** Stability evaluations of formulated MS-Hu6 at 100 mg/mL at 4°C and 25 °C for 90 days. (**A**) Samples were tested for thermostability using the protein thermal shift assay. At 4 °C, there was no noticeable $T_m$ shift for the Fab domain compared to day 0. At 25 °C, there is a slight left–shift in the $T_m$s of both the Fab and Fc domains at day 15, followed by relatively good stability up to 90 days for Fab. However, after 90 days of incubation, there is a loss of thermostability for the Fc domain with a $\Delta T_m$ of –1.51 °C. (**B**) Samples were tested for monomer loss using size exclusion chromatography (SEC). After 90 days, all three batches at 4 °C retained an average monomeric area of 99.4%, while the average monomer area retained at 25 °C was 96.8%, which is still within the acceptable limit of >95%.

The online version of this article includes the following source data for figure 6:

**Source data 1.** Source data for *Figure 6*.

## FSH binding studies

We evaluated formulated MS-Hu6 for its binding to human FSH. MS-Hu6 (1 μg/μL) in formulation buffer or PBS was incubated with and without 10 μg/μL FSH, and its thermostability was measured by protein thermal shift, as before. We expected that binding of FSH will stabilize the Fab, but not the Fc domain of MS-Hu6, and hence result in a right–shift of the $T_m$. We noted a $\Delta T_m$ of 4.49 ± 0.03 °C for MS-Hu6 in PBS and 4.80 ± 0.07°C for formulated MS-Hu6 (*Figure 7*). There was little or no change in $T_m$ in the Fc domain. These data establish conclusively the retained binding of formulated MS-Hu6 with its ligand, FSH.

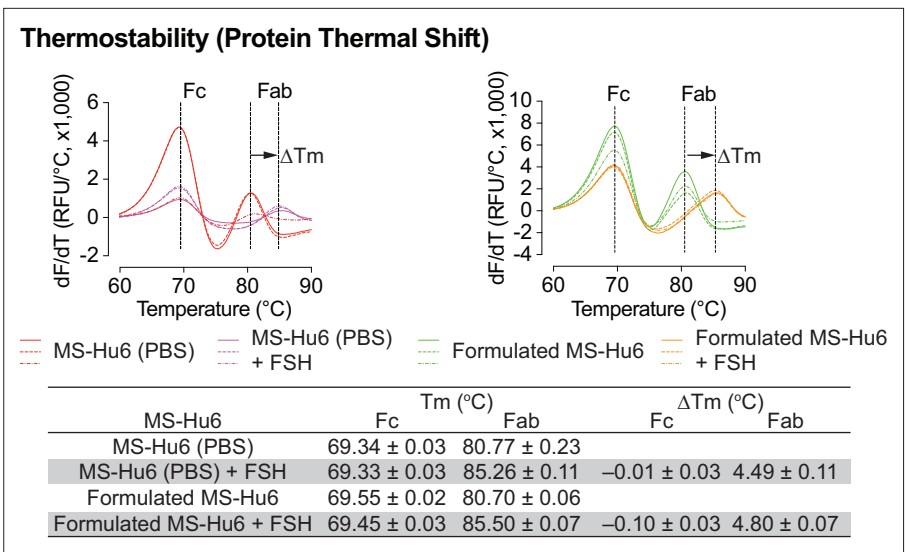

**Figure 7.** FSH binding of formulated MS-Hu6. FSH binding to MS-Hu6 in PBS and formulated MS-Hu6 is shown by a right shift in Fab $T_m$, which documents higher thermostability of the FSH:Fab complex.

The online version of this article includes the following source data for figure 7:

**Source data 1.** Source data for *Figure 7*.

## Discussion

We report the development and biophysical characterization of a formulation containing an ultra–high concentration MS-Hu6. We initially utilized protein thermal shift to find that formulated MS-Hu6 (0.2–100 mg/mL) had comparable $T_m$s when compared with MS-Hu6 in PBS with $\Delta T_m$ shifts of ≤1 °C-- this confirmed thermostability for both Fc and Fab domains compatible with industry standards (*Gera et al., 2022*; *Sant et al., 2023*; *He et al., 2010*). The improved thermostability is attributed to the stabilizing effect of excipients, such as sucrose, NaCl, Tween 20, L-methionine and disodium EDTA, each of which have distinct mechanisms of action.

To assess for monomeric stability and potential aggregation, we used SEC to find that the monomeric peak was >99% at all MS-Hu6 concentrations—this suggested that formulated MS-Hu6 was monomerically stabile at higher concentrations with minimal aggregation and no fragmentation. Clinical–grade antibodies must display active protein loss of not more than 5% and secondary species <2% in a freshly reconstituted therapeutic formulations (*Whitaker et al., 2017*; *NHS, 2017*; *Wang et al., 2007*). We complemented SEC with DLS to ascertain aggregate formation and colloidal stability. At each MS-Hu6 concentration in formulation, the major peak volume was >99% with a rh of 3–6 nm, and peak volumes for aggregated particles of <1%; this is consistent with that reported for several clinical–grade monoclonal antibodies (*Akbas et al., 2018*; *Li et al., 2011*). Of note is that soluble reversible aggregates up to 10 μm are accepted in biotherapeutics development. However, due to their potential immunogenicity or adverse reactions, particulate matter or large aggregates may need to be monitored under strict FDA guidelines (*Pham and Meng, 2020*; *Wang et al., 2007*).

Methionine oxidation of Fc domains is a common post–translational modification that can affect antibody bioactivity and potentially induce an immunogenic response. The interface of the $C_H2$ and $C_H3$ domains of most IgG1 antibodies contains two conserved heavy chain methionine residues. The oxidation of these methionine residues reduces thermal and colloidal stability (*Houde et al., 2010*; *Liu et al., 2008*), and in doing so, impairs long-term storage stability (shelf–life). The anti-oxidant L-methionine and chelating agent, disodium-EDTA are thus used to increase the shelf-life of commercial IgG1 biotherapeutics. Free L-methionine protects methionine residues from oxidation, while EDTA increases the effectiveness of anti-oxidants through its ability to complex heavy metal (*Zhou et al., 2012*; *Dion et al., 2018*). We found that the addition of 1 mM L-methionine and 1 mM disodium–EDTA added together resulted in thermostability in the protein thermal shift assay, and colloidal stability in DLS, with a major peak of 99.9% and rh 6.93 nm.

Viscosity is an essential parameter for any injectable dosage form, as highly viscous formulations increase back pressure and pain at injection sites. This issue is magnified for subcutaneously administered therapeutic antibodies at volumes <1.5 mL, making viscosity control a key determinant of highly concentrated formulations. We found an expected increase in viscosity with increasing concentrations of MS-Hu6, likely resulting from antibody crowding (*Raut and Kalonia, 2016*), compounded at high concentrations, by protein–protein interactions. However, the viscosity at 100 mg/mL MS-Hu6 was considerably less than the industry maximum of 50 cPs (*Tomar et al., 2016*; *Zhang and Liu, 2017*). The latter is known to affect syringeability. However, viscosity can vary between 30 and 50 cPs even at high antibody concentrations (>100 mg/mL). In recent years, viscosity control has been extensively studied as a function of concentration, temperature, pH, ionic strength, and ion type (*Tomar et al., 2016*).

Turbidity and clarity are equally important parameters in biopharmaceutical development. Increases in turbidity and reduced clarity could result from antibody overcrowding or gelation and the formation of reversible aggregates due to weak protein–protein interactions, such as Fc–Fc, Fc–Fab, hydrogen bonding, and hydrophobic interactions. We found a small increase in turbidity (absorbance) of 100 mg/mL formulated MS-Hu6 at 350 nm. *Per* European Pharmacopoeia, this would be categorized as Ref II (slightly opalescent ≤6 NTU). In the case of clarity, percent transmittance of formulated MS-Hu6 was >99% at all wavelengths, indicative of clarity between 0.2 and 50 mg/mL, with a slight decrease at 100 mg/mL (350 nm). The clarity values fell into the categories of Ref I (3–6 NTU) and II. The majority of commercialized biopharmaceuticals have clarity values up to 12 NTU (*Kingsbury et al., 2020*).

Modification of secondary structure is a sign of protein inactivation or denaturation. In agreement with other IgG1 therapeutics, CD spectroscopy revealed that the predominant secondary structures in MS-Hu6 are β-sheets and unordered/random coils (*Kelly et al., 2005*; *Moreno et al., 2016*; *Vermeer et al., 1998*). These structures remained unaltered in the formulated version with no loss of active protein. Active protein loss in a newly reconstituted biotherapeutic formulations should be <5%, with secondary species at a maximum of 2% (*Whitaker et al., 2017*; *NHS, 2017*; *Wang et al., 2007*). Intact secondary structure was confirmed by FTIR, in which the amide I band peak at 1637 cm$^{-1}$ (representing intra-molecular β-sheets) and the random coil structure peak at 1642–1657 cm$^{-1}$ were unaltered in formulated MS-Hu6 (100 mg/mL) compared with MS-Hu6 in PBS. The data rule out significant inter-molecular β-sheets resulting from protein aggregation or clustering. For further confirmation, we used the nano DSC to study multidomain transitions, as a surrogate for the conformation alterations of the Fab, Fc (C$_H$2), and Fc (C$_H$3) domains. Comparison between 50 and 100 mg/mL MS-Hu6 revealed almost overlapping peaks—further confirming retained conformational stability of the respective domains.

Aggregation can occur at various stages of product development, including freeze–thaw, manufacturing, filling, shipping, and storage (*Jain et al., 2021*). This could destabilize and degrade the product, and thereby reduce shelf-life. We thus carried out a rapid freeze–thaw (F/T) study at –80 °C/25 °C and –80 °C/37 °C and for three cycles to confirm formulation stability at extreme conditions. ΔT$_m$s were found not to be significant (<1 °C) for both Fc and Fab domains in all stressed conditions, indicating that formulated MS-Hu6 was thermostable under extreme stress conditions. The data also establish the absence of aggregation. For confirmation, the samples were further evaluated by DLS for colloidal stability. All samples showed excellent colloidal stability, with minimal reversible aggregation (<0.4%, acceptable limit <5–10) (*Kaur, 2021*). Moreover, the PDI or rh under stressed conditions were broadly comparable for three F/T cycles, further confirming colloidal stability. However, a small increase in PDI was noted at the third F/T cycle at –80 °C/37 °C, likely attributable to reversible aggregates due to Fc–Fc, Fc–Fab, and air–liquid interface interactions (*Raut and Kalonia, 2016*; *Kaur, 2021*; *Le Guyader et al., 2020*; *Torrente-López et al., 2022*). For three cycles, these reversible aggregates remained in the 0.06 to 0.14% range, which is within acceptable limits (5–10%). This finding suggests that the ultra–high MS-Hu6 formulation retains its hydrodynamic radius at extreme storage conditions (–80 °C/37 °C for three cycles).

Accelerated stability refers to the extent to which a drug retains, within specified limits, the near same properties and characteristics that it possessed at the time of manufacturing. It is a surrogate for shelf life and determines optimum storage conditions. Both Fab and Fc domains of formulated MS-Hu6 remained stable at 4 °C for 90 days (ΔT$_m$ <1 °C on thermal shift). Moreover, at 25 °C, the Fc domain was stable for 60 days (ΔT$_m$ <1 °C); however, after 90 days, the T$_m$ fell slightly (>1 °C),

indicative of reduced stability. In contrast, the $T_m$ for of the Fab domain did not change significantly ($\Delta T_m$–0.47 °C). SEC was used to further confirm monomeric stability after 90 days. At 4 and 25 °C, all three batches retained >99% and 96.8% monomer, with loss of <1% and 3.16%, respectively (acceptable limit for active structure loss is 5%). No fragmentation was noted in any samples at 4 °C (90 days). However, monomer loss was noted after 90 days at 25 °C, due to the formation of soluble dimers, with molecular weights greater than 150 kDa; the latter are not considered critical due to their reversible and transient nature. Such dimerization is induced by antibody crowding in solution, weak interactions between Fc–Fc, Fc–Fab, and Fab–Fab, and hydrophobic and air–liquid interactions (*Wang et al., 2007*; *Lu et al., 2013*; *Plath et al., 2016*). These soluble reversible aggregates are difficult to remove during antibody manufacturing and are therefore accepted in biotherapeutics (5–10%). However, and importantly, no fragmentation (low molecular weight species <150 kDa) was detected in any accelerated condition. Finally, MS-Hu6 in PBS and formulated MS-Hu6 were evaluated for their ability to bind FSH. Thermal shift assay revealed a right–shift for the Fab domain ($\Delta T_m$ = 4.49 ± 0.11 °C and 4.80±0.07 °C, respectively). Expectedly, and as we have shown before, the Fc domain in either instance remained relatively unperturbed ($\Delta T_m$– 0.1 °C) (*Gera et al., 2022*; *Sant et al., 2023*). However, while the ultra-high formulation exhibited higher thermostability compared to the low-concentration formulation (*Sant et al., 2023*), it is notable that, in addition to a higher MS-Hu6 concentration the former also had two additional excipients.

In summary, we reported development of an ultra-high concentration formulation for MS-Hu6, and document its thermal, monomeric and colloidal stability, structural integrity, stability under extreme and accelerated conditions, and effective binding to the ligand, FSH. The formulation is designed as a liquid solution formulation for a biologic, which is highly desirable in the therapeutic market. However, ensuring protein stability in a liquid solution is critical to biopharmaceutical development. To achieve this, we thoroughly screened various excipients to identify the composition that can most effectively stabilize the protein in the liquid solution system. Each excipient plays a crucial role in enhancing stability. For example, maintaining a pH away from the isoelectric point (pI) of the monoclonal antibody is essential to improve viscosity and thermostability. The ideal pH range is typically between 4 and 8, and a combination of buffers was used to achieve a pH of 6.2. Buffers are important for stabilizing antibodies and preserving the protein's conformation. The inclusion of surfactants, which are amphiphilic in nature, namely Tween 20, helped reduce interfacial tension and enhances stability. Sugars or polyols, such as sucrose, act as bulking agents, reducing freeze–induced degradation and promoting protein conformation stability. They also increase the unfolding temperature of the protein. The addition of salts can influence protein solubility and conformational stability, with their effect dependent on concentration. Of all these, from our perspective, antioxidant and chelating agents were particularly important as they help mitigate methionine and catalytic oxidation. Their protective effects significantly contribute to the long-term stability of our formulation.

## Materials and methods
### Materials
MS-Hu6 (Lot No. RO210015845) was provide by Genscript. Sodium chloride, potassium chloride and Tween 20 were purchased from Fisher Bioreagents, USA. Sodium dihydrogen phosphate, disodium hydrogen phosphate, and sucrose were from Alfa Aesar. L-Methionine and ethylenediamine-tetra-acetic acid (EDTA) (disodium salt) were procured from Sigma. Cell-culture grade water was obtained Corning. FSH (Cat # TLIH-A1110) was purchased from Golden West Biosolutions.

### Preparation of MS-Hu6 stock
All buffers were prepared in milli-Q water (Millipore) and filtered through 0.2 µm nitrocellulose filters (Millipore). The antibody was buffer-exchanged and concentrated by centrifugation at 3000 rpm and 4 °C in a Legend X1R Centrifuge (Sorvall) using Amicon Ultra-15 centrifugal filter units with a 30 kDa molecular weight cut–off (Millipore, Cat # UFC903024). To obtain the required concentration, the antibody was diluted serially with the formulation buffer. Concentration of MS-Hu6 before and after buffer exchange was measured by the absorbance at 280 nm using a Nanodrop ND-1000 Spectrophotometer (Thermo Fisher Scientific).

## Optimizing formulated MS-Hu6 at ultra-high concentration

The formulation for MS-Hu6 at ultra–high concentration was optimized using several FDA-approved excipients, used within the required range *per* the Inactive Ingredient Limits Guidelines (*Data.Gov, 2022*). Our final optimized formulation contains 20 mM phosphate buffer (pH 6.2), 260 mM sucrose, 1 mM sodium chloride, 0.001 % w/v Tween 20, 1 mM L-methionine and 1 mM disodium EDTA. In the formulation development process, we also employed a terminal sterilization method. This consisted of sterile filtration using a 0.22 μm sterile filter. The filtration process was carried out aseptically in a clean–air environment, and the formulation was filled and stored in sterile glass vials. This step was crucial in reducing aggregated particles and minimizing the risk of microbial contamination.

## Size exclusion chromatography (SEC)

SEC was carried out to detect monomer loss and aggregate formation (dimer, trimer, and multimer) using an AKTA Pure Fast Performance Liquid Chromatography System (Cytiva). Prepacked SEC columns (Superdex 200 10/300 GL 1×30 cm, particle diameter 13 μm) with TSKgel guard column SwXL (6 mm× 40 mm) were used. MS-Hu6 samples were diluted to 1 mg/mL using the formulation buffer, loaded onto the SEC column (500 μL), and eluted isocratically at a flow rate of 0.4 mL/min. The mobile phase is composed of sodium phosphate buffer at pH 6.2, 260 mM sucrose, 0.001% w/v Tween 20, and 1 mM NaCl at 25 °C. Protein concentration was measured at 280 nm. The area under absorption curve of the chromatogram was used to determine monomeric loss. The experiment was repeated twice, and representative chromatographs are reported.

## Dynamic light scattering (DLS)

To provide information on colloidal stability of formulated MS-Hu6, DLS was carried out to determine the size (hydrodynamic radius, rh) and homogeneity (polydispersity Index, PDI). Briefly, 20 μL formulated MS-Hu6 was diluted into 1 mL using formulation buffer or PBS and transferred to disposable 1 mL microcuvettes (Malvern Cat. ZEN0040). Using Zetasizer Nano-ZS 90 system (Malvern), the diluted sample was analyzed for 60 s at 90° scattering angle at 25 °C (5 cycles). The refractive index of the medium was set at 1.33 and dynamic viscosities were measured. Data were reported as Z–average of rh and PDI. All experiments were performed in duplicate, and representative particle size distribution (PSD) graphs are reported.

## Protein thermal shift assay

The protein thermal shift assay was routinely used to assess the thermostability of formulated MS-Hu6. The assay utilizes Sypro Orange (Cat # 01288948, ThermoFisher), which reports hydrophobic domains that are exposed during protein unfolding. Briefly, the 20 μL reaction mixture included 2.5 μL Sypro Orange dye (10 X), 0.58 μL antibody samples and 16.92 μL protein thermal shift buffer (i.e. formulation buffer). The protein samples were heated from 25 °C to 95 °C at a rate of 0.3 °C/s in a StepOnePlus real-time PCR system (Applied Biosystems), and fluorescence was captured in the ROX channel. All experiments were performed in triplicate for each concentration. $T_m$ was calculated based on the inflection point of the protein thermal shift melting curve. The thermal shift ($\Delta T_m$) between sample A and sample B was calculated using *Equation 1*:

$$\Delta T_m = T_m A - T_m B \tag{1}$$

## Viscosity measurement

A total of 100–500 μL of samples, including formulated MS-Hu6 at 1, 10, 20, 50, and 100 mg/mL, were analyzed using a Spindle S12 Viscometer (Rotavisc lo-vi, IKA) at 100 rpm and 25 °C to obtain dynamic viscosity values (Centipoise, Cp).

## Rapid freeze–thaw studies

Biotherapeutics are expected to experience extreme temperature conditions while being transported or stored. Freeze–thaw cycle testing is a type of stability testing that determines the robustness of an optimized formulation. Three batches of formulated MS-Hu6 (100 μL at 100 mg/mL) were subject to repeat freeze–thaws at –80 °C/25 °C or –80 °C/37 °C. Briefly, 100 μL formulated MS-Hu6 was incubated at –80 °C for 3 hr (TSX series freezer, Thermo Fisher). Samples were thawed for 30 min at 25 °C

in an Isotemp 250 thermostat (Fisher Scientific) or at 37 °C in a water bath (PolyScience), and analyzed by dynamic light scattering or protein thermal shift assay. The protocol was repeated to confirm antibody stability for up to three freeze-thaw cycles.

## Turbidity and clarity

Turbidity was measured by the attenuation of incident light due to scattering (*Wang et al., 2017*). At $OD_{350}$, while protein molecules have no intrinsic absorbance, aggregated particles could scatter light and thus generate opalescence. Clarity was evaluated based on percent transmitted light. Briefly, 50 µL protein samples were transferred into an uncoated flat bottom 96–well plate (Cat # 12565501, Fisher Scientific) and absorbance at 350 nm (for turbidity) or transmittance at 350 nm, 540 nm, and 595 nm (for clarity) was measured using a CLARIOstar microplate reader (BMG) at 25 °C after background subtraction (buffer without protein). All experiments were performed in duplicate.

## Fourier transform infrared spectroscopy (FTIR)

FTIR was carried out to confirm the structural integrity of formulated MS-Hu6 at high concentration. Our Nicolet iS10 FTIR Spectrometer (ThermoFisher) is equipped with an attenuated total reflectance (ATR) sampling accessory and a 45° ZnSe crystal. Briefly, 10–20 µL blank reference buffer or protein samples were loaded onto the ATR sample holder, and spectra were recorded from 500 $cm^{-1}$ to 4500 $cm^{-1}$. After subtracting the reference spectra from the sample spectra, smoothened second derivatives were generated using the Savitsky–Golay 7–point, 3rd order polynomial, 2nd derivative algorithm (Essential FTIR software, Operant). For appropriate peak fitting, the 2nd derivative peaks were multiplied by –1. The region of structural interest, Amide I, was used to assess structural integrity.

## Far-UV circular dichroism (CD) spectroscopy

Far-UV circular dichroism (CD) spectroscopy was used to assess secondary structure stability and integrity of formulated MS-Hu6. For this, 0.5 mg/mL (100 µL) of antibody samples were transferred to a cuvette. CD spectra were collected at 25 °C in a 10 mm path length with a 50 nm/min scan speed and a 0.45 s response time using a Chirascan CD spectrometer (Applied Photophysics). A bandwidth of 2 nm was used to average five scans from 180 to 250 nm at 25 °C. Built-in functions were used to subtract the reference spectra from the respective sample spectra, followed by smoothening using the Savitzky-Golay function. *Equations 2; 3* were used to compute the molar Ellipticity [Θ] and Mean Residue Ellipticity [MRE]:

$$[\theta] = \frac{\theta(\text{mdeg})}{10 * [\text{residues}](M) * l(\text{cm})} \tag{2}$$

$$[\text{Residues}](M) = [\text{Protein}](M) * n \tag{3}$$

where Θ is the measured/observed ellipticity (in millidegrees) at a particular wavelength, $l$ is the cuvette's path length (in cm), [protein] is protein concentration, and n is the number of amino acids in each protein under consideration. DichroWeb, created in the Wallace lab at the University of London, was used to process the secondary structure data (*Miles et al., 2022*). Data analysis was conducted using Contin-LL (Provencher & Glockner Method) and reference set *Miles et al., 2022*.

## Nano differential scanning calorimetry (Nano-DSC)

In addition to protein thermal shift assays, DSC was also performed to evaluate the thermal stability of formulated MS-Hu6 at high concentrations. Briefly, samples were diluted with formulation buffer to 5 mg/mL, and 500 µL of buffer (reference) or diluted sample were loaded into 24 K gold cylindrical capillary cells in a Model-602000 Nano DSC (Waters), equilibrated for 10 min, and heated from 25 °C to 90 °C at a rate of 1 ° C/min at a constant pressure of 3 atm. To obtain the precise unfolding temperature, the specific heat capacity (Cp) of the references were subtracted from the respective protein samples followed by data fitting using the three-transition model (NanoAnalyze Software). The respective thermograms for formulated MS-Hu6 is reported.

## Accelerated stability

Protein thermal shift assays and size exclusion chromatography were used to examine the stability of formulated MS-Hu6 at 4 °C (long term storage) or 25 °C (accelerated storage). Three batches for each

condition were evaluated. Samples were stored in 0.5 mL Eppendorf tubes and collected at 0, 15, 30, and 90 days. Protein thermal shift assays were performed to assess thermal stability. To determine monomeric stability, size exclusion chromatography was carried out on samples that have been stored for 90 days at 4 ° C and 25 ° C. Experiments were run in triplicate and representative chromatographs are reported.

## FSH binding assay

To confirm binding, formulated MS-Hu6 or MS-Hu6 in PBS (1 µg/µL) was incubated with or without human FSH (10 µg/µL) at room temperature for 30 min. Protein thermal shift assays using Sypro Orange were performed as above. Experiments were carried out in triplicate.

## Acknowledgements

Work at Icahn School of Medicine at Mount Sinai carried at the Center for Translational Medicine and Pharmacology was supported by R01 AG071870 to M.Z., T.Y., and S.-M.K.; R01 AG074092 and U01AG073148 to T.Y. and M.Z.; U19 AG060917 to M.Z. and C.J.R.; and R01 DK113627 to M.Z. M.Z. also thanks the Harrington Discovery Institute for the Innovator–Scholar Award toward the development of anti-FSH antibody. C.J.R. acknowledges support from the NIH (P20 GM121301). We also thank the Departments of Chemistry at Columbia University and New York University for their help with CD spectroscopic and Nano DSC analysis.

## Additional information

### Competing interests

Satish Rojekar: Is a co-inventor on a pending patent application relating to the ultra-high formulation of MS-Hu6. These patents are owned by Icahn School of Medicine at Mount Sinai (ISMMS), and the inventors and co-inventors. would be recipients of royalties, per institutional policy. Se-Min Kim, Daria Lizneva: Reviewing editor, *eLife*. Tony Yuen, Mone Zaidi: Senior editor, *eLife*. The other authors declare that no competing interests exist.

### Funding

| Funder | Grant reference number | Author |
|---|---|---|
| National Institute on Aging | R01 AG071870 | Se-Min Kim Tony Yuen Mone Zaidi |
| National Institute on Aging | R01 AG074092 | Tony Yuen Mone Zaidi |
| National Institute on Aging | U01AG073148 | Tony Yuen Mone Zaidi |
| National Institute on Aging | U19 AG060917 | Clifford J Rosen Mone Zaidi |
| National Institute of Diabetes and Digestive and Kidney Diseases | R01 DK113627 | Mone Zaidi |
| National Institute of General Medical Sciences | P20 GM121301 | Clifford J Rosen |

The funders had no role in study design, data collection and interpretation, or the decision to submit the work for publication.

### Author contributions

Satish Rojekar, Conceptualization, Data curation, Formal analysis, Validation, Investigation, Methodology, Writing – original draft; Anusha R Pallapati, Validation, Investigation, Methodology; Judit Gimenez-Roig, Marcia Meseck, Validation, Investigation; Funda Korkmaz, Data curation, Investigation;

Farhath Sultana, Clement M Haeck, Investigation, Methodology; Damini Sant, Data curation, Formal analysis, Investigation, Methodology; Anne Macdonald, Validation, Project administration; Se-Min Kim, Data curation, Funding acquisition; Clifford J Rosen, Conceptualization, Funding acquisition, Writing – review and editing; Orly Barak, Formal analysis, Validation, Project administration; John Caminis, Supervision, Validation; Daria Lizneva, Data curation, Validation; Tony Yuen, Formal analysis, Supervision, Funding acquisition, Methodology, Writing – original draft, Writing – review and editing; Mone Zaidi, Conceptualization, Supervision, Funding acquisition, Writing – original draft, Writing – review and editing

### Author ORCIDs
Satish Rojekar  http://orcid.org/0000-0002-3301-5941
Clement M Haeck  http://orcid.org/0000-0003-4856-1440
Clifford J Rosen  http://orcid.org/0000-0003-3436-8199
Mone Zaidi  http://orcid.org/0000-0001-5911-9522

### Decision letter and Author response
Decision letter https://doi.org/10.7554/eLife.88898.sa1
Author response https://doi.org/10.7554/eLife.88898.sa2

## Additional files

### Supplementary files
• MDAR checklist

### Data availability
All data generated or analysed during this study are included in the manuscript and supporting file; Source Data files have been provided for Figures 1-7.

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
