## [Editor Report]

This development of a highly concentrated and potentially clinically valuable antibody formulation for MS-Hu6, a first-in-class FSH-blocking humanized antibody is of potential translational importance in the management of osteoporosis, obesity, and Alzheimer's disease. The meticulous methodology is thorough and compelling in its range of techniques examining the stability and physiochemical properties of the formulated MS-Hu6.

---

## [Decision Letter]

**Decision letter after peer review:**

Thank you for submitting your article "Development and Biophysical Characterization of a Humanized FSH-Blocking Monoclonal Antibody Therapeutic Formulated at an Ultra-High Concentration" for consideration by *eLife*. Your article has been reviewed by 2 peer reviewers, and the evaluation has been overseen by a Reviewing Editor and Carlos Isales as the Senior Editor.

The reviewers have discussed their reviews with one another, and the Reviewing Editor has drafted this to help you prepare a revised submission. The reviewers consider the study to be well conducted, accurately justified, and with correct scientific reasoning. They consider the manuscript to be well-written and the experiments well-planned and designed. They target a few specific issues that need to be addressed.

Essential revisions:

*Reviewer #2 (Recommendations for the authors):*

1) To confirm FSH binding, formulated MS-HU6 was incubated with FSH for protein thermal shift assays in Figure 7. It is not clear what formulation (concentration) of MS-HU6 was used for the study. Is FSH binding similar for MS-HU6 formulations at higher concentrations (100 mg/ml) versus lower concentrations (1 mg/ml)?

2) Have the authors compared the potency of formulated versus non-formulated MS-HU6 preparations in an in vivo animal model?

3) The authors are encouraged to discuss what components of formulation or properties evaluated in this study can be considered novel.

---

## [Author Response]

Essential revisions:Reviewer #2 (Recommendations for the authors):1) To confirm FSH binding, formulated MS-HU6 was incubated with FSH for protein thermal shift assays in Figure 7. It is not clear what formulation (concentration) of MS-HU6 was used for the study. Is FSH binding similar for MS-HU6 formulations at higher concentrations (100 mg/ml) versus lower concentrations (1 mg/ml)?

Thank you for your comment. In the FSH binding assay, we utilized the formulated MS-Hu6 at a 100 mg/mL concentration. We serially diluted the formulation stock to perform the assay to achieve a final 1 µg/µL concentration. The diluted MS-Hu6 formulation (1 µg/µL) was then incubated with FSH at a concentration of 10 µg/µL for 30 minutes to ensure effective binding. Following the incubation, the mixture was analyzed using protein thermal assay to evaluate its thermostability. Notably, the binding of FSH to the Fab domain of MS-Hu6 was enhanced with the higher concentration of the modified stable formulation—with ΔT_m_ for the Fab domain of 4.49 ^o^C. In our previous study, we conducted the FSH binding assay at a lower concentration (0.15 mg/mL) (https://doi.org/10.1111/nyas.14952), which showed of ΔT_m_ of 2.68 ^o^C. However, while the ultra–high formulation exhibited higher thermostability compared to the low–concentration formulation, it is notable that, in addition to a higher MS-Hu6 concentration the former also had two additional excipients. This has been discussed in the manuscript on page 17, lines 403 to 405, of the revised manuscript.

2) Have the authors compared the potency of formulated versus non-formulated MS-HU6 preparations in an in vivo animal model?

Thank you for your comment. As of now, we have not conducted potency studies of this formulation in our animal models. However, it is part of our future plan to evaluate this formulation's potency and pharmacokinetic parameters. We recognize the importance of assessing its efficacy and determining its performance in vivo, and we intend to carry out these studies to further validate the potential of our formulation.

3) The authors are encouraged to discuss what components of formulation or properties evaluated in this study can be considered novel.

Thank you for your comments. This formulation is designed as a liquid solution formulation for biologics, which is highly desirable in the market. However, ensuring protein stability in a liquid solution is critical to biopharmaceutical development. To achieve this, we thoroughly screened various excipients to identify the most stable composition that can effectively stabilize the protein in the liquid solution system. Each excipient plays a crucial role in enhancing stability. For example, maintaining a pH away from the isoelectric point (pI) of the monoclonal antibody (mAb) is essential to optimize viscosity and thermostability. The ideal pH range is typically between 4 and 8, and a combination of buffers is used to achieve this. Buffers are important for stabilizing antibodies and preserving the protein's conformation. The inclusion of surfactants, which are amphiphilic in nature, helps reduce interfacial tension and enhances stability. Sugars or polyols act as bulking agents, reducing freeze–induced degradation and promoting protein conformation stability. They also increase the unfolding temperature of the protein. The addition of salts can influence protein solubility and conformational stability, with their effect dependent on concentration. Of all of these, from our perspective, antioxidant and chelating agents are particularly important as they help mitigate methionine and catalytic oxidation. Their protective effects significantly contribute to the long-term stability of the formulation. We have discussed this in closing on pages 17 to 18, lines 406 to 424, of the revised manuscript.